# Targeting Youths' Intentions to Avoid Food Waste: Segmenting for Better Policymaking

Ewelina M. Marek-Andrzejewska * and Anna Wielicka-Regulska

Department of Economics and Economic Policy in Agribusiness, Faculty of Economics, Poznań University of Life Sciences, 60-624 Poznań, Poland; wielicka@up.poznan.pl
* Correspondence: ewelina.marek@up.poznan.pl

**Abstract:** Food waste is a global challenge that raises many questions about the reasons and prevalence of this phenomenon in all sectors of the economy. The youth is regarded as a consumer group, which is the most prone to food waste. This paper aims to understand their food waste intentions to support tailored policies for policymakers, retailers, and other market actors. We applied the extended Theory of Planned Behavior (TPB) to find the relevant variables that affect the youth's intention not to waste food. Besides creating a general model, we divided the sample into segments differing in respondents' intentions to avoid food waste and specific socioeconomic characteristics. The data confirm significant differences between young women and men from urban and rural areas. Each of the segments was characterized by specific latent variables, influencing the intentions to avoid food waste. This segmentation allowed for developing policy recommendations that were tailored to each segment. It is a unique approach to differentiate the youth to unveil their specific food-waste intentions. Based on the above, we conclude that segmenting is a useful approach to the general TPB model, allowing for interesting insights. A fine segmentation is also a milestone to develop tailored policies, interventions, and communication on food waste reduction in rural and urban areas.

**Keywords:** consumer attitudes; Millennials; gender; rural areas; urban areas

## 1. Introduction

Food waste is a global challenge that raises many questions about the reasons and prevalence of this phenomenon in all sectors of the economy. The topic has become one of the Sustainable Development Goals of the United Nations, which calls for halving per capita global food waste at retail and consumer levels by 2030 [1]. According to FAO [2], around 1.3 billion tons of food are lost or wasted worldwide, equivalent to one-third of food produced for human consumption. In the European Union, approximately 88 million tons of food are wasted [3]. The highest share of food waste is observed at the household level and was estimated at 53 percent of the total food waste [3]. Between 158 and 298 kg per person per year of food was wasted in the European Union (EU) in 2006 [4]. The top five of the most profligate countries in the European Union constituted Germany (10.3 million tonnes), the Netherlands (9.4 million), France (9 million), and Poland (8.9 million) in 2006 [4].

Past research reveals that different socioeconomic groups showcase different attitudes towards food waste and perform various behavior types in this regard [5]. The youth (particularly university students) establish one of the most food-profligate groups in developed countries [6–8]. Thus, this group should be of particular interest to researchers and policymakers and should be mainly targeted by policy interventions [7]. Young people display a full spectrum of food waste behaviors ranging from ostentatious food-wasting [9] to participation and advocacy in the zero-waste movement [10–12]. Studies in Romania and South Korea indicate that this group is not homogeneous [9,13] and requires in-depth research to formulate actionable recommendations for policymakers.

Although many studies focus on cognitivism and follow waste-related intention and behaviors based on the Theory of Planned Behavior (TPB) model [14–17], the usefulness of consumer segmentation based on the TPB model has not been proved yet, to the best of the authors' knowledge. Moreover, a segmentation of youth representing one of the most profligating age groups has not been reported. Thus, detailed policy recommendations, taking into account this group's heterogeneity, have been developed. Our study investigated the usefulness of the TPB model in the cluster analysis of youth, proving its efficacy in segmentation and delivering parsimonious policy interventions.

For our research, we developed the general model, based on the Theory of Planned Behavior (TPB) (thereby adding additional variables) for consumers (mainly the youth) who lived in the northwest part of Poland (i.e., Great Poland). In addition to classical elements of the TPB model, we identified several critical additional factors in the literature that were also considered in building and testing our extended TPB model. More precisely, our model mirrors the TPB model and its extension proposed by Visschers et al. [18] that will be explained in this paper. To maintain the comparability with the methodology used by Visschers et al. [18], food waste was defined as food that is thrown away to waste or biowaste bin, or can be composted or become the feed to animals. Such a description is in line with the definition of the United Nations [1].

Subsequently, we segmented them according to respondents' intentions to avoid food waste and by socioeconomic variables. More precisely, to gain additional insights into youths' intentions to prevent food waste, we divided our sample into three segments, characterized by specific latent variables affecting intention to avoid food waste. Our method revealed that these latent variables depend on gender, age, body mass index, educational level, place of residence, household size, and household income. The results obtained were compared to other studies. As an outcome, we created three specific youth segments and then developed tailored policy interventions for each of them. By doing so, we were able to contribute to the knowledge gap on addressing food waste.

The paper is composed as follows. Section 2 presents a conceptual framework. Section 3 describes the methodology. Results are presented in Section 4, which is followed by a discussion in Section 5. Section 6 offers policy implications and recommendations. Section 7 concludes this paper.

## 2. Conceptual Framework

### 2.1. Psychological and Behavioral Predictors of Intention to Avoid Food Waste

Although the usefulness of TPB is sometimes questioned, e.g., [19], it is still regarded as the dominant and efficient quantitative model in understanding the intentions and the vast extent of consumer behaviors in an environment-related domain [20]. The basic constructs of the TPB model are attitudes, subjective norms, and perceived behavioral control. These three constructs predict an intention and behavior in question.

In the academic literature, it has been suggested that the predictive power of the basic TPB model can be enhanced by adding other relevant and empirically tested constructs [14,15]. Our study was based on a model developed and verified by Visschers et al. [18]. Thus, the TPB model was extended by the following constructs: knowledge of expiration dates, knowledge of food storage, household planning habits, and the identity of a good host and a good provider. The inclusion of these variables improved the TPB model's capacity noticeably to predict intention and behavior related to food waste either in a study of Visschers et al. [18] and in the current study.

Personal attitudes. Personal attitudes towards food waste are an essential determinant of intentions to avoid food waste in quantitative and qualitative studies. The survey reflecting the TPB model among younger and older consumers revealed that personal attitudes constitute a major predictor of an intention to avoid food waste [18,21,22] in TPB-based surveys. Recent studies indicate that consumers (including the youth) were negative about food waste and felt guilty throwing the food away [22]. Nevertheless, consumers considered that they wasted less food than they did [23,24]. They were attaching importance not to

waste food, reduced food waste, and affected intentions to reduce food loss at the household level [18]. Considering the above, we developed the following hypothesis for our study:

**Hypothesis 1 (H1).** *Positive personal attitudes towards food waste are positively correlated with a higher level of intentions to avoid food waste.*

Past research indicates that apart from personal attitudes, three other types of attitudes directly link to the intention to avoid food waste, namely financial, environmental, and health attitudes.

Financial attitudes. Financial concerns were mentioned in many studies as having a greater impact on intentions and behaviors that reduce food waste than environmental or social motives [14,25–27]. In qualitative [15] and quantitative [22] studies, the financial motive related to wasting money was of crucial importance in increasing intentions not to waste food. One of the main problems was overpurchasing [28]. Furthermore, price-conscious consumers could waste less food [6,21,29,30]. Therefore, our next hypothesis was formulated as follows:

**Hypothesis 2 (H2).** *Positive financial attitudes towards food waste are positively correlated with a higher level of intentions to avoid food waste.*

Perceived health risks. Health motives can be of great importance in decisions about food use. In Qi and Roe's [24] study, almost 70 percent of respondents agreed that throwing food away could reduce the risks of foodborne diseases. Consumers, who were aware of expiration dates or monitored the length of time the food had been stored at home, were found to throw away more food [26,31]. It should be noted that labeling food was not always clear to consumers, and knowledge in this area could significantly reduce food waste [32,33]. Healthy attitudes also concern healthy diets. On the one hand, by eating healthy, consumers reduce their food waste [28]. On the other hand, health-conscious consumers tended to purchase too much varied food for their family members that was not all eaten [15,34]. Thus, our next hypothesis was defined as:

**Hypothesis 3 (H3).** *Positive health attitudes towards food waste are positively correlated with a higher level of intentions to avoid food waste.*

Subjective norms. In the TPB, subjective norms were the weakest predictor of behavioral intention in several areas [35]. The same conclusions were drawn by researchers focusing on food waste, i.e., [15,16,18], which could explain that food waste behaviors were not visible to others. Nevertheless, we included the latent variable on social norms into our study design and tested it in hypothesis 5:

**Hypothesis 4 (H4).** *Subjective norms on food waste are positively associated with intentions to avoid food waste.*

Personal norms. Typical studies targeted at individuals employ "personal norms" or "moral norms" construct into the TPB model that measures consumer morality linked to the food waste problem. They can be recorded threefold: by declarations in quantitative [14,18], or qualitative studies [36,37], or assessed by a third party (usually a researcher) through the observation of routines and rituals performed in households [38,39]. According to past studies, respondents often felt guilty after throwing food away and showed or declared a responsibility to avoid food waste or generally to reduce the environmental footprint [26,28,40]. Personal norms added to the TPB model in previous studies on proenvironmental behavior explained consumer behavior and reduced the model's unexplained variance [41]. Thus, our hypothesis regarding the role of personal norms follows:

**Hypothesis 5 (H5).** *Personal norms on food waste are positively correlated with intentions to avoid food waste.*

Perceived behavioral control (PBC). PBC as an antecedent influencing intention to limit food waste and the same behavior proved significant in many studies [14–16,18,21]. However, it is worth noting that not always perceived behavioral control has a significant impact on reducing food waste. The study by Thompson et al. [32] showed that in the case of behaviors related to dairy waste and knowledge of expiration dates, PBC had no significant impact on intentions. Factors that limit PBC may be the pickiness of other household members, the irregularity of meals at home, the need to use large food packages, or perceived difficulty preparing meals from food leftovers [17]. In this context, our hypothesis was formulated as follows:

**Hypothesis 6 (H6).** *The higher the lack of perceived behavioral control, the lower the intention to avoid food waste.*

Knowledge of expiration dates and food storage (KEDES). On the one hand, KEDES has been proven to be additional motivators or facilitators in reducing food waste [27,42]. On the other hand, a lack of understanding of the difference in the terminology of expiration dates contributes to greater food waste [32]. Given the above, we hypothesized that:

**Hypothesis 7 (H7).** *A good level of knowledge on expiration dates is positively correlated with the intention to avoid food waste.*

**Hypothesis 8 (H8).** *A good level of knowledge of food storage is positively correlated with the intention to avoid food waste.*

Identity of a good host and a good provider. Another feature in the study was the identity of a good host and a good provider. It is associated with offering an abundance of fresh food and a variety that usually leads to more food waste [15,18,34,43]. As a result, we hypothesized that:

**Hypothesis 9 (H9).** *The identity of being a good provider is positively associated with the intention to avoid food waste.*

Household planning habits. Research indicates a significant impact of food consumption management (FCM) habits, mostly shopping and planning habits in households. These habits have proven to be very important for reducing food waste among low- and high-income households [16,18,44–46]. Therefore, our hypothesis was stated as follows:

**Hypothesis 10 (H10).** *Household planning habits are positively correlated with the intention to avoid food waste.*

Intention to avoid food waste. The high motivation to reduce or prevent food waste is linked to less food waste [18,47]. Studies indicate that PBC and attitudes towards food waste [15,47] have the greatest influence on intentions.

In addition to the TPB's essential components, our research also took into account knowledge of expiration dates and food storage, household planning habits, and the good host's identity, and the good provider. We followed here the research assumptions of Visschers et al. [18].

To sum up, Figure 1 presents our hypotheses and the extended TPB model for the general model and three segments. Apart from the main hypotheses, we also assumed that there exist positive covariances between the latent variables.

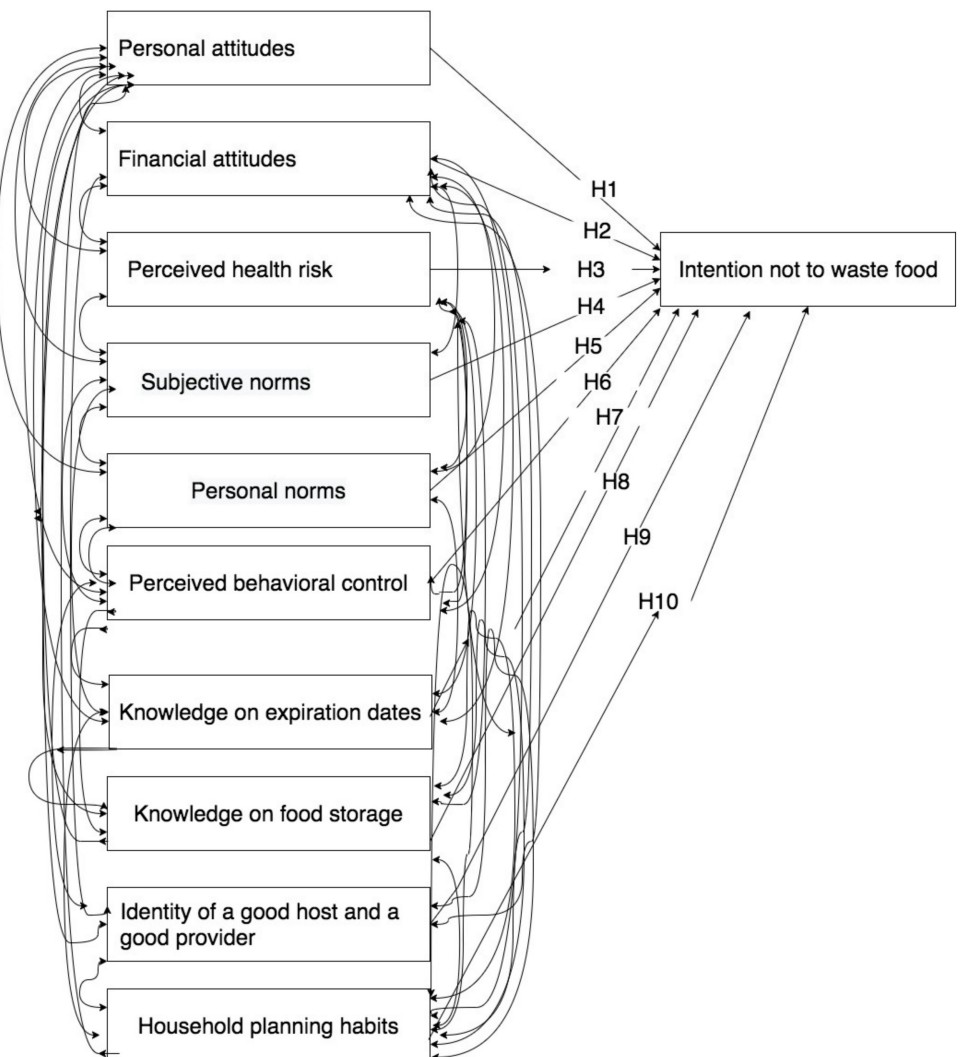

**Figure 1.** Hypotheses and the extended Theory of Planned Behavior (TPB) model.

*2.2. Socioeconomic Predictors of Intention to Avoid Food Waste*

As already mentioned, we hypothesized that the intention to avoid food waste is also directly related to socioeconomic variables, and the general TPB model does not fully reflect the consumer's intentions to avoid food waste. Therefore, we sought to differentiate our sample by respondents' socioeconomic status and its impact on the intention to avoid food waste. We split the sample into smaller segments, thereby considering gender, age, BMI, educational level, place of residence, household size, and household income.

Gender. Gender seems to be the most discursive primary sociodemographic variable that potentially affects the intention to limit food waste. Most studies indicate that men waste more food than women [8,43,48], although the study of Koivupuro et al. [49] pointed to higher food waste in women. Thus, the relationship between gender and food waste may differ between countries [49].

Age. Studies show that age was negatively correlated with food wasted. Older people managed food more rationally [6,16,18,22,43]. The elderly were more likely to plan their purchases and have more knowledge about food wastage [23,28] or may have experienced food shortages during World War II, as Visschers et al. [18] suggested. By contrast, young consumers profligated a lot and were considered the most food-wasting group in developed countries [6–8].



Body mass index (BMI). Raghunathan et al. [50] found consumers' positive association between food-waste-aversion and BMI. Contrarily, Banna et al. [51] did not find this confirmation in adolescent girls. The study of Mallinson et al. [52] also did not confirm the relationship between food waste and BMI in all identified consumer segments. Due to past mixed results, BMI was included in our study.

Educational level and employment. Higher education significantly increases the likelihood of food wasting behavior [8]. The more years spent on education, the higher the probability of food waste generation. The majority of studies explain this relationship by increasing income experienced by better-trained people or by the inability of less-educated people to self-assess the amount of food wasted [8]. Likewise, those who are not employed or work part-time waste less food than those who work full-time [8].

Place of residence. According to most studies, rural households waste less than their urban counterparts [53,54]. For example, the Norwegian rural households throw approximately 36 kg/cap/year of food away [53] and urban households—42.1 kg/cap/year [55]. However, some quantitative studies (based on the EU data) indicate that households from urban and rural areas do not differ in terms of food waste [7]. Place of residence can also determine attitudes and thus drive different behavior regarding food waste [56].

Household size and structure. Studies usually indicate that the larger the household, the higher the food waste [43,49]. Larger households usually include generally children. Their presence meant greater food waste than households without dependent children [18,25,42]. The age of children may have also influenced the level of food waste. Younger children attending primary school usually wasted more food than older children [57].

Household income. The material situation of a household also affects the level of food waste. Greater freedom to spend money in high-income households leads to buying unnecessary food and wasting it [7,25,54,58,59]. Furthermore, in high-income households, their members may have limited time and mental resources to prevent and avoid food waste [6]. Contrarily, students' financial constraints force them to buy lower quality or perishable foodstuff to irregular provisioning and eating behaviors that impede thoughtful planning [7]. Money spent on groceries is also related to the amount of food that is wasted. The more money spent on groceries, the more food waste a household generates [26].

## 3. Methodology

### 3.1. Measurement of Constructs

This paper draws on survey findings, which included questions needed to build the extended Theory of Planned Behavior (TPB) model regarding food waste among youth in Poland, specifically in its voivodship Great Poland. It is the third voivodship in terms of population (3.5 million) and gross domestic product (GDP = 13 million EUR) in Poland. The GDP is twice the Polish average and the largest in Poland, after Warsaw. The region's capital is Poznań, inhabited by around 536,000 persons.

The questionnaire was conducted online via the Survey Monkey tool from 15 June to 31 October 2018. The survey was filled out by respondents (mainly students) who received a link to the questionnaire and were asked to forward it to other friends. It is based on the Visschers et al. [18] survey, who kindly agreed to work on their questions. It was adapted to meet the reality of Polish consumers. The questionnaire encompassed questions that were grouped into many latent variables. Questions were built on a five-point Likert scale; higher scores corresponded to a higher agreement with the statement—each latent variable aimed at responding to the research hypotheses presented in the conceptual framework.

Among 563 respondents, 369 filled out the survey completely and were between 20 and 34 years old, and only those responses were considered for further analysis. There were 267 female and 102 male respondents. An average respondent was 24 years old, and his BMI was equal to 22.82 (normal weight) (according to Table 1). He lived in a large city with three persons in his household. He obtained a secondary education degree and studied at a large public university. An average net household income was equal to 4000–5000 PLN (around 936-1170 EUR).

**Table 1.** Descriptive statistics of the sample.

| | Median | Mean | Std. Deviation | Skewness | | Kurtosis | |
|---|---|---|---|---|---|---|---|
| | | Statistic | Statistic | Statistic | Std. Error | Statistic | Std. Error |
| Gender | 1.00 | 1.28 | 0.448 | 0.996 | 0.120 | −1.013 | 0.239 |
| Age | 24.00 | 24.93 | 8.352 | 2.068 | 0.120 | 4.888 | 0.239 |
| Household income | 5.00 | 5.40 | 2.847 | 0.535 | 0.120 | −0.653 | 0.239 |
| Educational level | 6.00 | 5.44 | 0.891 | −1.388 | 0.120 | 0.914 | 0.239 |
| Number of persons per household | 3.00 | 3.38 | 1.421 | 0.236 | 0.120 | −0.775 | 0.239 |
| Place of residence | 1.00 | 2.29 | 1.598 | 0.706 | 0.120 | −1.209 | 0.239 |
| BMI (body mass index) | 22.30 | 22.820 | 3.5968 | 1.258 | 0.120 | 2.651 | 0.239 |

*3.2. Data Analyses*

The data were analyzed in two software for statistics: STATA and SPSS, using the maximum likelihood method for confirmatory factor analysis (CFA). We applied structural equation modeling (SEM) and covariances between latent variables, which had an impact or not on the primary latent variable "Intention not to waste food" in the TPB model. Although the survey was built on the 5-point Likert scale, the data were treated as continuous and not ordinary. According to Schumacker and Beyerlein [60], Mîndrilă [61], and Babakus et al. [62], when the sample size is large enough ($N > 200$), ordinal data have a large number of categories (minimum five). When they are approximately normal, the maximum likelihood method does not produce biased results. These arguments convinced us to treat the data as continuous when computing the correlation and covariance matrix and model fit estimate for CFA called RMSEA (i.e., root mean square error of approximation).

The data were subsequently analyzed in SPSS, based on the component factor analysis and k-means factor analysis. The results of the Kaiser–Meyer–Olkin measure of sampling adequacy were equal to 0.74 (chi-square = 453.717). They were satisfactory to conclude that the component factor analysis was appropriate for the data set and to estimate the most important factors influencing the latent variable ("Intention not to waste food"). The component factor analysis was performed by using the maximum likelihood analysis as the extraction method. It aimed to identify the most critical variable among those describing the latent variable ("Intention not to waste food").

The component factor analysis results fed the k-means analysis. The variable "I always try to eat all the food I bought" was used to represent the "Intention not to waste food" for labeling the cases. This variable reached the highest scores for factor and score coefficient with chi-square = 17.560 ($p < 0.000$). However, before the proper k-means analysis, the variables age, BMI, educational level, number of persons in a household, and household income were standardized with a mean of 0 and standard deviation equal to 1, making these scales comparable. Furthermore, the selected variables were tested for normality. According to the Shapiro–Wilk test, all of them were normally distributed at $p = 0.000$. The validity of this test suggested that the k-means cluster analysis could be used for data segmenting. In the next step, the k-means analysis was run to define four clusters describing the respondents. All variables were statistically significant in the ANOVA table, and the number of respondents in each cluster was well distributed.

## 4. Results

*4.1. Descriptive Results*

Questionnaire items were statements to which respondents had to express either agreement or disagreement on a five-point Likert scale, higher scores corresponded to a higher agreement with the statement. Their means and standard deviations are presented in Appendix A. In most cases, the means oscillated between 3.50 and 4.00,

meaning that respondents usually agreed with the statements. An exception concerns questions on perceived behavioral control, in which case the questions were formulated negatively, and thus, respondents mainly choose 2 and 3. Besides, questions grouped under the item "Perceived health risks" oscillated around 3.00, meaning that respondents were relatively neutral in responding to questions on the linkage between health and food consumption. For the data analyses, questionnaire items were appropriately grouped into latent variables, characterized by a reasonable to high-reliability scale, measured via Cronbach's α. The alpha coefficient for all items was between 0.57 and 0.86. Therefore, we included all items in the analyses.

*4.2. General Model*

First, the CFA was estimated based on the maximum likelihood method without considering respondents' sociodemographic aspects. We included all latent variables in the confirmatory factor analysis in STATA. The RMSEA was slightly above the acceptable level of 0.06. Therefore, we excluded a latent variable called "Subjective norms," which was not statistically significant in the model until the RMSEA reached the level of 0.051, which was assumed to be a good model fit. The fact that "Subjective norms" were excluded from the model is in line with other researchers' results. Armitage and Conner's meta-analysis [13] also revealed that the variable "Subjective norms" was usually a weak predictor of intentions in other studies' TPB model. In this study, the impact of "Subjective norms" was also minor.

Another not statistically significant variable was "Financial attitude," but it had to be included in the model to achieve RMSEA at 0.051. The fact that this variable was not correlated with the "Intention not to waste food" (abbreviated as "Intention") suggests that generally, the youth was not concerned about the financial aspects of food waste.

Figure 2 presents that the essential latent variable ("Intention not to waste food") was influencing "Personal attitude" (r = 0.71, *p* < 0.001), meaning that one's attitude regarding food waste played an essential role in building the behavioral intention. Such a result is similar to previous findings, e.g., [18,22].

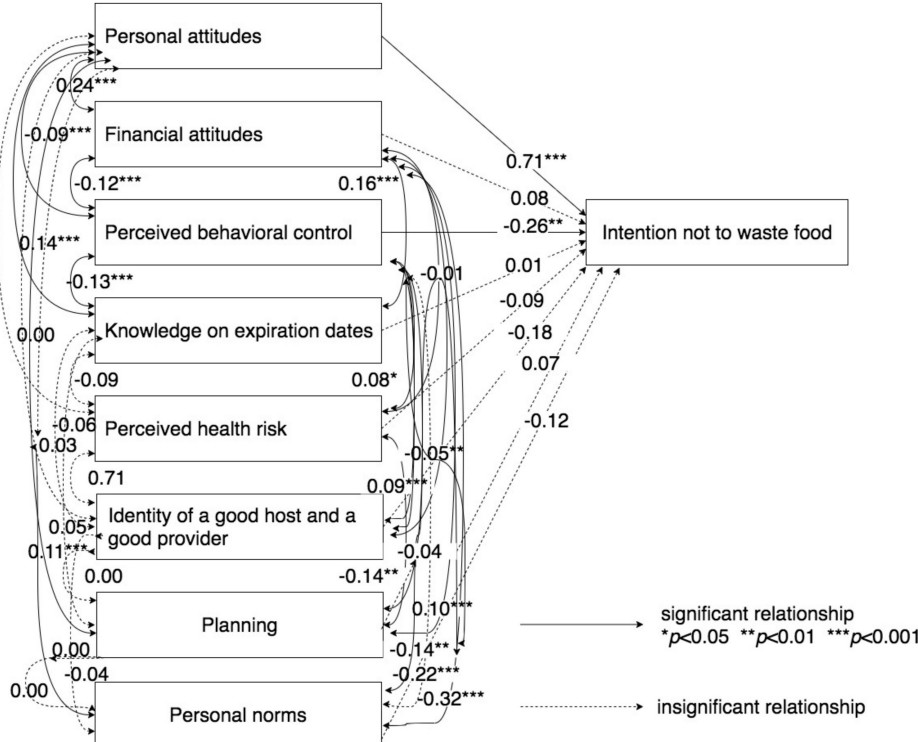

**Figure 2.** The intention not to waste food in the general model. Full lines mean significant relationship; dotted lines mean insignificant relationship; * *p* < 0.05; ** *p* < 0.01; *** *p* < 0.001.

In addition, "Perceived behavioral control" was also negatively correlated to "Intention" at r = −0.26, $p < 0.01$. This result indicates that subjects (who expressed an intention not to waste food by consuming all purchased foodstuff ($\overline{\chi}$ = 4.16) and by generating little food waste ($\overline{\chi}$ = 3.77) did not consider it to be challenging to ensure that only small amounts of foodstuff were thrown away in their households ($\overline{\chi}$ = 2.68) and to plan shopping in such a way in that all purchased items were consumed by household members ($\overline{\chi}$ = 2.58). Besides, respondents were positive regarding their ability to prepare a new meal from food leftovers ($\overline{\chi}$ = 2.86). These results were similar to those obtained by Graham-Rowe et al. [15], Mondéjar-Jiménez et al. [21], Stefan et al. [16], Visschers et al. [18], and Werf et al. [14].

The remaining variables were not statistically significant in the model, indicating no impact on "Intention not to waste food." Nevertheless, a few of these variables were correlated with other latent variables. For example, "Personal attitude" was positively correlated with "Financial attitude" (r = 0.24, $p < 0.001$), "Knowledge on expiration dates" (r = 0.14, $p < 0.001$), and "Planning" (r = 0.11, $p < 0.001$).

Considering the first correlation, subjects who thought "food waste" as "immoral" ($\overline{\chi}$ = 4.02) also considered "food waste" as a synonym of "money waste" ($\overline{\chi}$ = 4.52). They stated that it was possible to find ways to avoid food waste ($\overline{\chi}$ = 3.83) and got nervous when the food was wasted in their household ($\overline{\chi}$ = 3.96). Moreover, they admitted that they could not afford to throw away foodstuff ($\overline{\chi}$ = 3.87) and disagreed with the statement that they seldom think about the money when they throw the food out ($\overline{\chi}$ = 2.52).

The second correlation was weak, and it indicated that respondents' positive personal attitude to avoid food waste was linked to a lack of knowledge on expiration dates. They were not convinced that there were differences between the expiration dates ($\overline{\chi}$ = 3.79). Not all of them knew that products with a "use-by" date should not be eaten after the expiration date ($\overline{\chi}$ = 3.77), nor that products with a "best-before" date could be eaten after the expiration date ($\overline{\chi}$ = 3.44). Moreover, they did not believe that the risk of eating the foodstuff, which passed the "use-by" date, was high ($\overline{\chi}$ = 3.08), nor that consumption of food, which passed the "best-before" date, was safe ($\overline{\chi}$ = 2.79).

The third correlation was also weak and was related to planning. In general, respondents stated that they liked to plan things ($\overline{\chi}$ = 3.77), but it was somewhat challenging to plan meals in their households and manage food leftovers ($\overline{\chi}$ = 2.89).

Among other latent variables, a strong negative correlation was found between "Perceived behavioral control" and "Personal norms" (r = −0.22, $p < 0.001$). This means that respondents felt bad when they had to throw the food away ($\overline{\chi}$ = 3.77) and tried to avoid it ($\overline{\chi}$ = 4.02). At the same time, they found it easy to ensure that only small amounts of food were thrown away in their households ($\overline{\chi}$ = 2.68).

"Personal norms" were also strongly correlated with "Financial attitude" (r = 0.32, $p < 0.001$). Thus, it can be concluded that respondents' moral obligations to avoid food waste were positively correlated with the financial aspects of food waste. In other words, those who considered throwing the food as throwing the money away ($\overline{\chi}$ = 4.52) also felt obliged to limit their food waste ($\overline{\chi}$ = 3.92).

*4.3. Segments' Characteristics and Models*

4.3.1. Segment 1: Control-Conscious Young Men from Urban Areas

Segment 1 encompassed men (N = 101), who were 24 years old on average, and their average BMI was equal to 24.49 (normal weight). They finished secondary school, were students, and lived in Poznań. The net monthly household income was about 4001–5000 PLN (around 936–1170 EUR), comparable to the sample's average. There were usually three persons per household. For segment 1, the TPB model was developed (reaching RMSEA at 0.051) (see Figure 3). It was necessary to include almost all latent variables in the model to achieve such a good fit to the model, even though most of the variables were not statistically significant concerning the "Intention" variable.

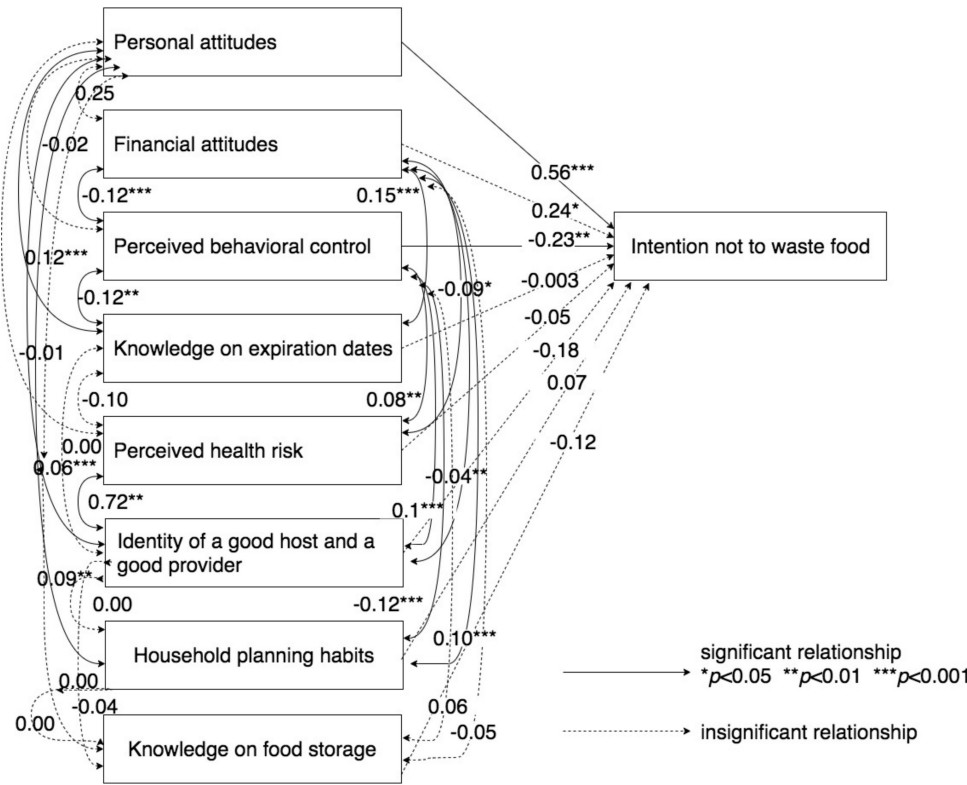

**Figure 3.** TPB model for segment 1. Full lines mean significant relationship; dotted lines mean insignificant relationship; * *p* < 0.05; ** *p* < 0.01; *** *p* < 0.001.

The highest correlation was found between the variables "Personal attitude" and "Intention not to waste food" (r = 0.56, *p* < 0.001). It was even higher than in the general model by 0.08. Whether a person tried to reduce food waste was thus very much related to the fact that a person had a positive attitude towards such behavior. Another critical variable that had an impact on the "Intention" variable was "Perceived behavioral control" (r = −0.23, *p* < 0.01), similar to the general model. It means that subjects from segment 1 did not find it challenging to prevent food waste in their households, and they thought they had enough control to avoid it. These results suggest that both "Personal attitude" and "Perceived behavioral control" were strong predictors of whether young men would have had an intention to avoid food waste. Unlike in the general model, "Financial attitude" was not statistically significant in the TPB model for segment 1 (r = 0.24, *p* > 0.05). Men were thus less concerned about the financial aspects of food waste in their households.

Other latent variables were not statistically significantly correlated to the "Intention" variable but were statistically significantly correlated with other latent variables. For example, "Personal attitude" was positively correlated to "Knowledge on expiration dates" (r = 0.12, *p* < 0.001), "Host identity" (r = 0.06, *p* < 0.001), and "Planning" (r = 0.09, *p* < 0.01). These correlations were weak but strongly statistically significant, implying that, for example, respondents who stated that "It is not necessary to waste food. It is always possible to use the food leftovers somehow" ($\overline{\chi}$ = 3.5) declared that they knew about the differences between the "best-before" and "use-by" dates ($\overline{\chi}$ = 3.8). Respondents of segment 1 as hosts were not so concerned when there was not enough food for their guests ($\overline{\chi}$ = 2.6), and they were not buying too much fresh foodstuff for their households ($\overline{\chi}$ = 2.6). They also stated that they liked planning ($\overline{\chi}$ = 3.7), but they did not plan very well their meals in advance ($\overline{\chi}$ = 2.7) and what to do with food leftovers ($\overline{\chi}$ = 2.8). About the "Perceived behavioral control" variable, statistically significant correlations were found to "Knowledge on expiration dates" (r = −0.12, *p* < 0.01), "Perceived health risk" (r = −0.08, *p* < 0.01), "Host identity" (r = −0.01, *p* < 0.001), and "Planning" (r = −0.12, *p* < 0.01). These correlations were relatively weak, however, suggesting a connection between the variables.

### 4.3.2. Segment 2: Positive-Attitude Young Women from Urban Areas

The majority of the survey's respondents (*N* = 164) were classified into segment 2. These were women aged 24. Their average BMI was below the sample's average and was equal to 21.90 (normal weight). They lived in Poznań and had a secondary educational level. At the time of the survey, they were students of a large public university. On average, 2–3 persons mainly inhabited their households. The net income level was above the sample's average (i.e., 5001–6000 PLN, so around 1170–1405 EUR). The TPB model for segment 3 was characterized by a good model fit at RMSEA equal to 0.063 and is presented in Figure 4.

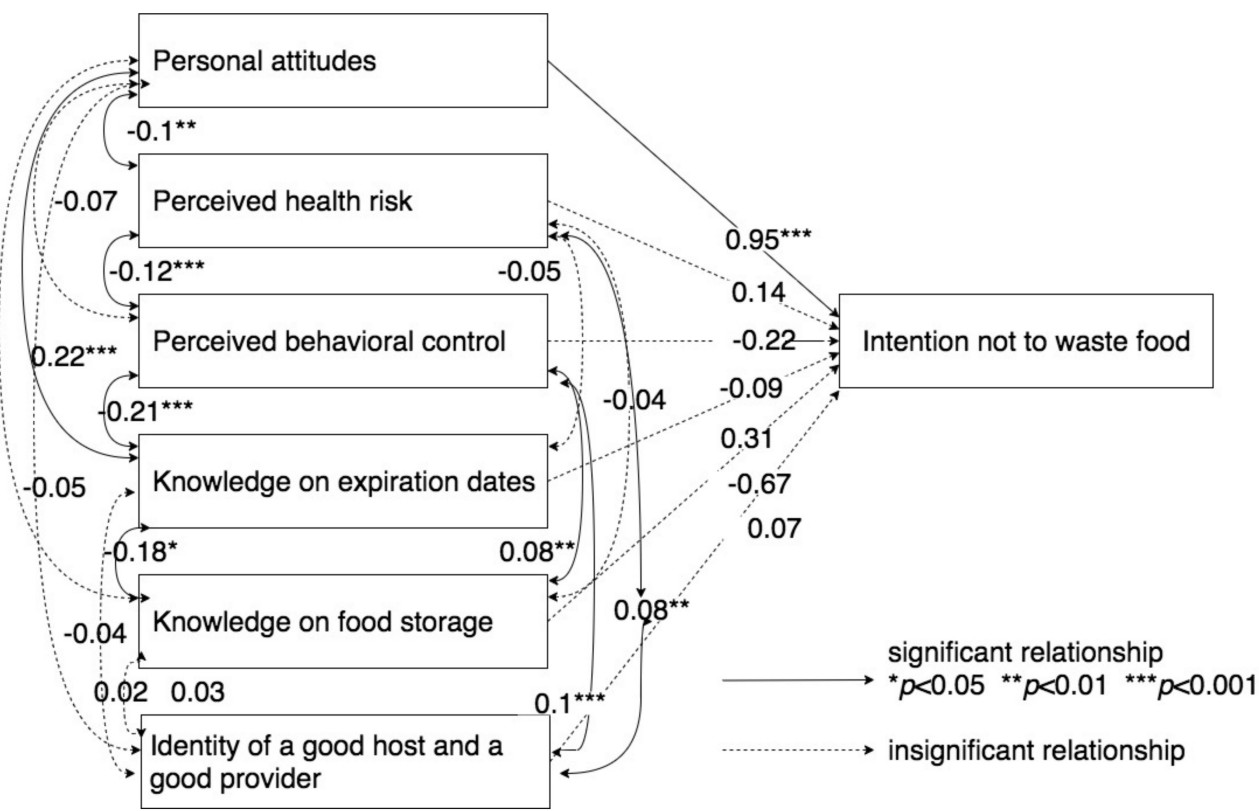

**Figure 4.** TPB model for segment 2. Full lines mean significant relationship; dotted lines mean insignificant relationship; * *p* < 0.05; ** *p* < 0.01; *** *p* < 0.001.

The model encompassed several latent variables, but only one of them was statistically significant, i.e., "Personal attitude" (r = 0.95, *p* < 0.001). Such a strong correlation to the "Intention not to waste food" variable results in equal medians for both variables. In other words, for all variables describing the "Personal attitude" and "Intention not to waste food," the responses were equal to four on the five-point Likert scale, except for "I try to use food leftovers in other dishes again." The average values were also very close to four, meaning that respondents agreed to the statements related to "Personal attitude" and "Intention not to waste food" variables.

The "Personal attitude" variable was further correlated to "Perceived health risk" (r = −0.1, *p* < 0.01) and "Knowledge on expiration dates" (r = 0.22, *p* < 0.001). Whereas the correlation with the former variable was weak, the other correlation was strong. The correlation between the last pair of variables indicates that respondents acknowledged that there were differences between the "best-before" and "use-by" dates ($\overline{\chi}$ = 3.75) and agreed to the statement that products after the "best-before" date are still eatable and safe for human health. Simultaneously, respondents of segment 3 were nervous when uneaten food was thrown away ($\overline{\chi}$ = 3.98) and found it immoral to throw food away ($\overline{\chi}$ = 4.10).

Furthermore, there were statistically significant correlations (but weak) to the "Perceived health risk" variable, including "Perceived behavioral control" (r = −0.12, $p < 0.001$) and "Host identity" (r = 0.08, $p < 0.01$). A stronger correlation was found between the "Perceived behavioral control" and "Knowledge on expiration dates" variables (r = −0.21, $p < 0.001$), suggesting that subjects, who did not consider it difficult to control the amounts of thrown away food in their households, knew about the differences between the expiration dates and agreed with a statement that food after the "best-before" date can be eaten. Similar strength of correlation was also found between the "Knowledge on expiration dates" and "Storage knowledge" variables (r = −0.18, $p < 0.001$), indicating that subjects who knew about the differences between the expiration dates also disagreed with statements that fruits and vegetables should not be stored separately ($\overline{\chi} = 2.43$) and that potatoes should be stored in a fridge ($\overline{\chi} = 3.80$).

4.3.3. Segment 3: Planning–Seeking Young Women from Rural Areas

Segment 3, similarly to segment 2, was represented by women (*N* = 104) but slightly younger, aged around 22. Their BMI was below the average, at approximately 21.32 (normal weight). The educational level was similar to segment 3 (i.e., secondary school level), currently studying at a large public university. They lived in rural areas and their household's net income corresponded to the average (4001–5000 PLN, equal to around 936–1170 EUR). These respondents lived in larger households than others, notably of four persons. This means that the household income per capita was lower than in other segments. The TPB model is presented in Figure 5 (RMSEA = 0.058).

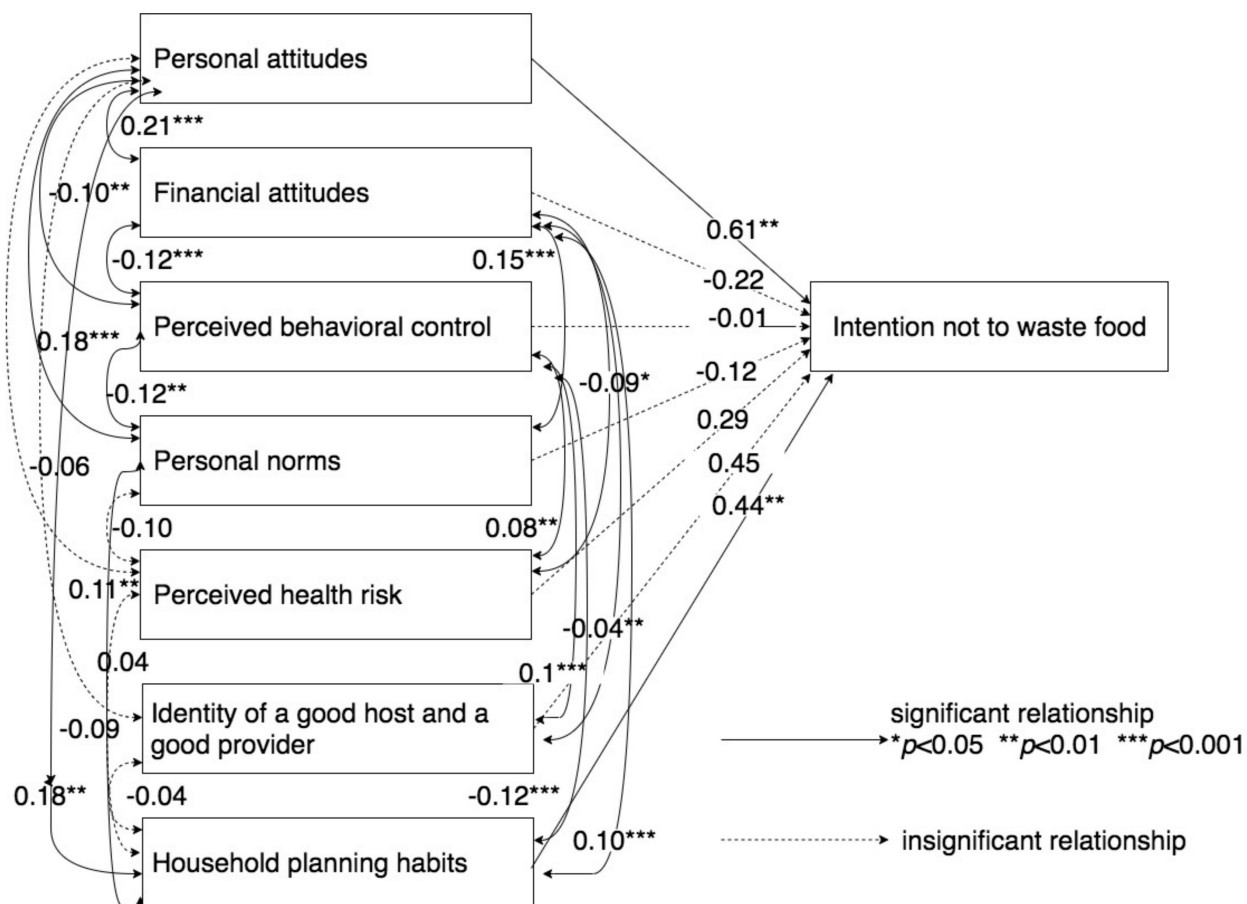

**Figure 5.** TPB model for segment 3. Full lines mean significant relationship; dotted lines mean insignificant relationship; * $p < 0.05$; ** $p < 0.01$; *** $p < 0.001$.

Similar to other models, the most crucial latent variable influencing the "Intention" was "Personal attitude" (r = 0.61, *p* < 0.01). For this group of respondents, it was essential not to throw away food because they considered it immoral ($\overline{\chi}$ = 4.14), probably due to embeddedness in the agricultural environment and agriculture through formal and informal bounds. At the same time, they tried to avoid food waste ($\overline{\chi}$ = 4.14). Unlike in other models, the "Intention not to waste food" variable was also statistically significantly correlated to "Planning" (r = 0.44, *p* < 0.01). The correlation was positive and strong, which means that respondents who liked planning ($\overline{\chi}$ = 4.00) also tried to avoid food waste in their households ($\overline{\chi}$ = 4.00). Nevertheless, they did not always stick to their plans regarding meals ($\overline{\chi}$ = 2.87) or stacked to grocery shopping lists ($\overline{\chi}$ = 2.78).

Moreover, the "Planning" variable was statistically significantly correlated to "Personal attitude" (r = 0.18, *p* < 0.01), "Financial attitude" (r = 0.10, *p* < 0.001), "Perceived behavioral control" (r = −0.04, *p* < 0.001), and "Personal norms" (r = 0.11, *p* < 0.01). Amongst the above pairs of latent variables, the strongest positive correlation was between "Planning" and "Personal attitude." This result strengthened the validity of the model for segment 4 in that the main two latent variables influencing the intention to avoid food waste were strongly positively correlated.

Furthermore, it should be noted that although the "Financial attitude" variable was not statistically significantly correlated to the "Intention" variable, it was correlated to all other variables in the model. The financial aspects of food waste were thus prominent in the TPB model for segment 3 and were particularly relevant for the "Personal attitude" and "Planning" variables. For these two latent variables, the correlations to the "Financial attitude" variable were strong and positive at r = 0.21 (*p* < 0.001) and r = 0.18 (*p* < 0.01), respectively. They were also statistically significantly correlated to the "Intention" variable. Such a result indicates that food waste's financial aspects shaped the personal attitude towards food waste and respondents' planning activities.

Moreover, they indirectly influenced their intention to avoid food waste. For example, an average respondent of segment 4 admitted that food waste meant money to waste ($\overline{\chi}$ = 4.63) and that she could not afford to pay for food, which would have been wasted afterward ($\overline{\chi}$ = 4.00). Simultaneously, she considered it immoral to waste food when people in other parts of the world were starving ($\overline{\chi}$ = 4.14) and got nervous when uneaten food was thrown away ($\overline{\chi}$ = 4.19). She was not a very good planner, though, in that she did not always consider what to do with food leftovers while preparing a meal ($\overline{\chi}$ = 3.10), nor planned meals in her household ($\overline{\chi}$ = 2.87). Nevertheless, she stated that she liked to make plans in general ($\overline{\chi}$ = 4.01).

This segment is also characterized by a high level of moral norms (called "Personal norms" in the model) that were correlated with most of the latent variables but were not correlated with the "Intention" variable. More precisely, young women from rural areas considered it immoral to throw food away, and their moral aspects were positively linked to "Personal attitude" and "Financial attitude" at r = 0.18 and r = 0.15, respectively. They were also negatively correlated with "Perceived behavioral control" at r = 0.12.

## 5. Discussion

In this study, we aimed at applying an extended version of the TPB model to young consumers on the topic of food waste avoidance via an online questionnaire. Overall, our findings reveal that the extended TPB can explain the intentions to avoid food waste among youth very well. Moreover, it was possible to develop the general model for the entire population sample and divide the sample into three segments according to specific characteristics related to the intention to avoid food waste.

In all presented models, "Personal attitude" was the most critical variable and had a statistically significant positive correlation with "Intention not to waste food." Similar conclusions were drawn in studies of Stancu et al. [22], Pakpour et al. [63], and Visschers et al. [18] on consumers, including the youth. The linkage between personal attitudes and intentions

seems to be independent of age and gender, and a positive attitude is crucial in preventing food waste.

Furthermore, other studies reveal that "Perceived behavioral control" (PBC) should be considered as an antecedent influencing intention to avoid food waste [14–16,18]. In our research, the PBC was also statistically significant in the general model and was particularly important to young men from urban areas, as indicated in segment 1. That said, there was no significant correlation between PBC and intentions to avoid food waste among young women. This result suggests that young men were more optimistic than the responding women regarding their household members' ability to ensure that only small amounts of food were wasted and plan grocery shopping to eat all purchased foodstuffs. Such a result is perhaps explainable because men are instead not responsible for shopping, cooking, cleaning up food leftovers, and other activities in their households. The lack of or a minor engagement in the above activities makes them optimistic about avoiding food waste in their households.

Moreover, we used the latent variable "Subjective norms" to evaluate it concerning the intention to avoid food waste. The variable was not statistically significant in either of the models. This result is in line with previous research of Graham-Rowe et al. [15,16,18] and research that was not related to food waste [35].

In addition, the "Financial attitude" variable was not correlated with "Intention not to waste food" in all models. Younger respondents, be it men or women, were less concerned about the financial implications of food waste. It can be concluded that younger respondents were less motivated to avoid food waste due to financial aspects. This can be explained by the fact that the financial burden of purchasing food is not visible to the young population because their parents are responsible for food purchases.

Additional variables to the traditional TPB model also played an essential role in the models presented, but they were usually not directly linked to the intention to avoid food waste. Instead, they had implications on other variables in the segments. An exception can be found regarding "Planning" that was strongly positively correlated with "Intention not to waste food" in segment 3, represented by young women from rural areas. Such a result implies that these women (more than other respondents) liked to plan things. They stated that although they wanted to make plans, they sometimes had difficulties planning meals and grocery shopping. It should be noted that women in segment 3 had a slightly lower income than the youth in other segments, and more persons lived in their households. These characteristics imply that careful planning activities were more prominent in their daily activities than among women or men from urban areas.

Furthermore, although all respondents were convinced that they knew the difference between the expiration dates, they generally did not provide correct responses to whether or not to eat food after the "best-before" or "use-by" date. Visschers et al. [18] also did not find this variable to be strongly correlated to the intention to avoid food waste. However, in our models, the variable "Knowledge on expiration dates" was rather weakly but statistically significantly correlated with other variables in the general model and segment 2, encompassing young women from urban areas. It seems that respondents linked the financial consciousness of throwing food away with their knowledge on expiration dates.

## 6. Policy Recommendations

Our study revealed that the youth should be considered in specific segments because each segment has different food waste approaches, and their intentions are linked to other latent variables. As a result, we propose to diversify policy tools to better target each of the segments. This section commences with general policy recommendations applicable to all segments and suggests specific recommendations for each segment.

### 6.1. General Recommendations

Generally, the youth has a positive personal attitude towards the intention to avoid food waste. This positive attitude should be further fostered by policymakers and other

market actors (e.g., retailers) to improve the youth's intention not to waste food and perform food-wasting behavior. Actions should strengthen the conviction that wasting food is not right and diminished could be the right way to reinforce positive behavior. It would be beneficial to communicate that purchasing and wasting less food is better than buying more oversized packages of food (e.g., "buy two and get one for free") or buying and cooking more than needed at a time. In other words, policymakers and other market actors should communicate new social norms of sufficiency, as was also proposed by Aschemann-Witzel et al. [5] to other consumers.

Young consumers should get a broader picture of the consequences related to food waste. It is needed that they will be aware of links and impacts regarding financial, environmental, moral, and social aspects. Youth seem to be less concerned about the financial impact of food waste (probably because they are usually not in charge of purchasing food while living with their parents). In order to make them more aware of monetary losses, they should be asked to contribute to the household budget or to do grocery shopping on behalf of their household. They should also be more engaged in the preparation of meals and the management of food leftovers. They could also participate in classes in which they have to simulate the management of a household budget. In these proposals, it is necessary to educate the youth and their parents about the financial aspects of food waste to prevent it.

Apart from educational campaigns or knowledge dissemination regarding household management, internet-savvy young consumers would benefit from mobile and computer apps that would support their management efforts via better planning. These applications would foster planning and management practices and help control the food-wasting behavior (for example, by delivering actionable tips on how they can better plan food purchases and how they can prepare some wholesome meals from food leftovers). Indeed, future households equipped with Internet-of-things devices will also support better household food management. The awareness of financial losses due to food waste should also be communicated as an economic benefit of avoiding waste. In other words, consumers should be aware of financial savings due to food waste avoidance in their households.

Similarly, young consumers should be better educated about the link between food waste and its impact on the environment. Therefore, while communicating on financial aspects, it would be advised to cover environmental issues as well. Young consumers should be aware of the amounts of water and other resources needed to produce food and how harmful food wastage is. They should be informed via educational campaigns about environmental issues. Additionally, they may be prone to dedicated logos or certificates highlighting market actors' involvement in food waste avoidance (analogically to logos on organic food, for example) [64].

Furthermore, the youth should be encouraged to plant their vegetables. Even in a small apartment, it is possible to grow small paprikas, chive, or beet sprouts on the windowsill. For example, retailers can expand their assortment by seeds of plants and sprouts. The youth may also want to purchase them spontaneously in their school canteens. They should also have opportunities to compost uneatable food either in a biowaste bin or in a dedicated place for composting. Municipalities, schools, and universities could encourage creating common composters in common gardens [65]. It would be beneficial to engage the youth in creating public vertical gardens and composters in cities and small villages [66]. Such places would help the youth to communicate with nature and better appreciate the natural environment. As a result, they should better recognize the linkage between food waste and the environment.

When drawing a broader picture of food waste consequences, moral issues should be highlighted. Our research revealed that only young women from rural areas felt guilty and immoral when throwing the food away. Moral norms could be strengthened by engaging the youth in food sharing or gleaning by NGOs or citizens, donating food to food banks [67], and by making the youth more aware of hunger in third countries, while in the developed countries so much food is being wasted. Furthermore, highlighting moral benefits (along

with financial aspects) of the right food selection and better management of meals could help to be more consequent about planning grocery shopping and meals, and to cut the food waste that can appear at the purchase stage or during the consumption.

*6.2. Policy Recommendations for Segment 1: Control-Conscious Young Men from Urban Areas*

Our research revealed that young men from urban areas were quite confident concerning personal behavioral control. The reason for that can be twofold: either they were indeed able to control food waste in their households or were not able to judge household food management adequately. Assuming the latter, young men should become more engaged in household chores to better assess the household's food waste. It would be beneficial to provide easy tips for cooking, storage, and food handling to reuse food leftovers. A mobile app would be helpful, so their parents or flatmate could play an essential role in this regard. Young men should be encouraged to participate in cooking events organized by governments, municipalities, schools, or universities. However, assuming that young men were indeed very good at personal behavioral control, they could be taken as examples. They could share their advice and knowledge with others (via social media or directly at a local level).

Young men's perceived behavioral control was also statistically significantly correlated with financial attitude and knowledge on expiration dates, suggesting that they were prone to financial aspects of food waste and knowledgeable of expiration dates. Therefore, market actors can offer lower prices on foodstuff, which is close to the expiration date or has become suboptimal in supermarkets or other grocery stores. They may also want to decrease prices of foods not conforming to current market standards, offer "happy hours" for food or beverages that need to be eaten fresh in restaurants or canteens. Moreover, the government or municipalities could support stores' opening with products that are suboptimal and are close to the expiration date. The support could be provided by amendments in the law, subsidies for a new opening, tax exemptions for such stores, etc.

*6.3. Policy Recommendations for Segment 2: Positive-Attitude Young Women from Urban Areas*

Our study reveals that the young female generation from urban regions is sensitive to the food waste problem and is keen to acquire knowledge to prevent it. Although the perceived behavioral control of young women from urban areas was not statistically significantly correlated with the "Intention not to waste food" variable, it was correlated with "Perceived health risk", "Knowledge on expiration dates" and "Storage knowledge". Therefore, it was concluded that young women from urban areas seemed to have a high level of food waste literacy. They could potentially serve as influencers in their communities and social media. They could share their knowledge about household food management with others and provide examples of good practices with their peers and other interested audiences. They should be encouraged to set up their blogs and social media channels to communicate and engage in citizen interaction on best household practices. They represent a strong positive attitude towards the intention to avoid food waste in their households. This attitude could be well used in spreading the knowledge of food waste. It is also proof that they are sensitive to the food waste problem.

At the same time, these young women from urban areas would undoubtedly appreciate the provision of easy tips for storage and food handling from public authorities or other bodies, such as nongovernmental organizations (NGOs). They could pass these pieces of information to others. They may be keen to participate in food sharing and gleaning initiatives and support other related activities in their cities, including cooking courses or trade fairs. With their ability to acquire new knowledge and influence others, they could become a part of the educational campaigns organized by public authorities or retailers. Besides, they may also pay attention to a unique logo showing market actors' involvement in food waste avoidance initiatives. They could help to spread the information about it. They could also serve as a medium in communicating new social norms on thriftiness and

simplicity. Providing this group with inspiration and more knowledge on relevant aspects of food waste reduction could cause beneficial spillover effects.

*6.4. Policy Recommendations for Segment 3: Planning–Seeking Young Women from Rural Areas*

Young women from rural areas were characterized by a lower income and more persons in their households, on average, compared to those in urban areas. They were also distinguished regarding their willingness to plan nonfood-related activities. Nevertheless, they admitted that planning other things put grocery shopping and meals aside and contributed to food waste in their households. From a policymaker's perspective, it should be considered to provide tools that simplify household food management. For example, they would benefit from a mobile application, which could help make shopping lists, monitor food waste, and provide tips for using food leftovers. Such an application would also help them set up food-related goals and simultaneously increase their perceived behavioral control over food waste behavior. It would facilitate shopping and meal planning in their households.

Furthermore, young women from rural areas considered it immoral to throw food away and took the financial aspects of food waste into account to a greater extent than other peers. Perhaps living outside the urban areas, they were more exposed to food poverty than the youth representing the remaining segments, so communicating that food waste reduction helps fight hunger and undernourishment could confirm the social or even personal relevance of diminishing food waste. Coherently, observation of food waste's financial dimension was statistically positively linked to the "Planning" variable. Young women from rural areas discerned the link between food waste and economic disadvantage, leading to food poverty. Stressing the financial consequences of food waste could lead to the consolidation of food waste, reducing behavior. Policymakers and retailers could engage in educational initiatives in rural areas that highlight food waste problems [68].

As financial aspects of food waste are particularly important for this segment, it would be advised that retailers reduce prices of foodstuff prices to become suboptimal or are close to the expiration date. It would be recommended to offer special corners with reduced articles at the point of sale to help consumers find special offers. It would also be useful to follow the Danish example of opening shops with suboptimal products or close to the expiration date at reduced prices. Restaurants and canteens could offer "happy hours" for meals and other food leftovers that were not eaten during the regular opening hours.

## 7. Limitations of the Study and Directions for Future Research

Our study enabled distinguishing specific segments among the youth, and proposed targeted policy recommendations for each of them. Nevertheless, our study has certain limitations that could be addressed in future research. First, our sample was overrepresented by women, but it should not be a major drawback as women are usually responsible for household food management. Moreover, our study could be expanded by including young men from rural areas as they were underrepresented, and this should be addressed in the future. Second, our model explains a considerable amount of the variance in the intentions to avoid food waste, but there is still the possibility of improvement. We recommend adding more control variables as routines, kitchen equipment, and organization to reflect better the automatic behaviors, ease of performing desirable behaviors, and situational cues appearing along the food waste journey. Third, a particular dimension connected to food consumption and waste in the social context assessed in this study was "an identity of a good provider." Still, the social life intensity was not reported, which could be addressed in future studies. Fourth, it seems plausible to test the youth's proposed policy interventions by surveying, observing, or in-depth interviewing combined with relevant experiments. Fifth, we can also expect differences among countries to arise from specific cultural and socioeconomic conditions. Thus, conducting similar research in other countries would be recommended for cross-country comparisons. Finally, the COVID-19 outbreak could have

modified the youth's intentions to avoid food waste [69], so further research in these new circumstances is advised.

## 8. Conclusions

Youth are perceived as one of the most wasteful consumer groups. They contribute significantly to food waste at the household level. Therefore, their food-related behavior and intentions to avoid it should be investigated with greater attention. There is also an urgent need to take actions that would influence them and lead to behavioral changes. Our methodology enabled the segmentation of this group of consumers. It was possible to distinguish slight differences between young women and men from both rural and urban areas. We identified specific latent variables, which influence their intentions to avoid food waste. Subsequently, we provided tailored policy recommendations for policymakers, retailers, and other market actors regarding food waste avoidance. Although our study has certain limitations and further research is needed, we could better target youth's intentions to avoid food waste. The differences presented should be triggered to deepen the research among other socioeconomic groups and acquire targeted knowledge to influence their food-waste intentions and behavior effectively.

**Author Contributions:** Conceptualization, E.M.M.-A.; methodology, E.M.M.-A., A.W.-R.; software, E.M.M.-A.; validation E.M.M.-A.; formal analysis, E.M.M.-A.; investigation, E.M.M.-A.; resources, E.M.M.-A. and A.W.-R.; data curation, E.M.M.-A.; writing—original draft preparation, E.M.M.-A. and A.W.-R.; writing—review and editing, E.M.M.-A.; visualization, E.M.M.-A.; supervision, E.M.M.-A.; project administration, E.M.M.-A. Both authors have read and agreed to the published version of the manuscript.

**Funding:** This work has been supported by the funds of the Ministry of Science and Higher Education granted to the Poznan University of Life Sciences.

**Institutional Review Board Statement:** Not applicable.

**Informed Consent Statement:** Not applicable.

**Data Availability Statement:** Not applicable.

**Acknowledgments:** The authors wish to express special thanks to Ewa Jerzyk from the Poznań University of Economics and Business and Aneta Disterheft from QIAGEN for their initial comments on the work proposal, and Elżbieta Goryńska-Goldmann for her help in data gathering.

**Conflicts of Interest:** The authors declare no conflict of interest.

## Appendix A

**Table A1.** Questionnaire items per construct, including the mean, standard deviation and internal reliability (Cronbach's $\alpha$).

| | Mean | Std. Deviation | Cronbach's $\alpha$ |
|---|---|---|---|
| | **Statistic** | **Statistic** | **Statistic** |
| **Intention to avoid food waste** | | | **0.76** |
| I try to waste no food at all. | 4.14 | 1.005 | |
| I always try to eat all purchased foods. | 4.16 | 0.941 | |
| I try to produce only very little food waste. | 3.76 | 1.096 | |
| I aim to use all leftovers. | 3.29 | 1.253 | |
| **Personal attitude** | | | **0.67** |
| It is unnecessary to waste food. It can always be used in some way. | 3.83 | 1.064 | |
| It is immoral to discard foods while other people in the world are starving. | 4.02 | 1.192 | |
| It upsets me when unused products end up in the waste bin. | 3.96 | 1.147 | |

<div align="center">**Table A1.** *Cont.*</div>

| | Mean | Std. Deviation | Cronbach's $\alpha$ |
|---|---|---|---|
| | **Statistic** | **Statistic** | **Statistic** |
| **Financial attitudes** | | | **0.58** |
| I think that wasting food is a waste of money. | 4.52 | 0.849 | |
| I cannot afford to pay for foods that are then discarded. | 3.87 | 1.108 | |
| Saving money does not motivate me to discard less food. | 2.67 | 1.319 | |
| I rarely think about money when I throw food away. | 2.52 | 1.359 | |
| **Perceived health risks** | | | **0.69** |
| I believe that the risk of becoming ill as a result of eating food past its use-by date is high. | 3.08 | 1.217 | |
| I am not worried that eating leftovers results in health damage. | 2.53 | 1.295 | |
| I think that consuming leftovers is harmless. | 2.92 | 1.274 | |
| I think that one can perfectly safely eat food products whose use-by dates expired a few days ago. | 2.78 | 1.269 | |
| **Subjective norms** | | | **0.62** |
| People who are important to me find my attempts to reduce the amount of food wasted unnecessary. | 2.47 | 1.203 | |
| People who are important to me are greedy when I try to reduce my food waste. | 1.93 | 1.041 | |
| **Personal norms** | | | **0.86** |
| I feel bad when I throw food away. | 4.26 | 1.011 | |
| I feel obliged not to waste any food. | 3.92 | 1.073 | |
| It is contrary to my principles when I have to discard food. | 3.81 | 1.139 | |
| I have been raised to believe that food should not be wasted and I still live according to this principle. | 4.02 | 1.071 | |
| **Perceived behavioral control** | | | **0.74** |
| I find it difficult to prepare a new meal from leftovers. | 2.86 | 1.227 | |
| I find it difficult to make sure that only small amounts of food are discarded in my household. | 2.67 | 1.162 | |
| I find it difficult to plan my food shopping in such a way that all the food I purchase is eaten. | 2.56 | 1.261 | |
| I have the feeling that I cannot do anything about the food wasted in my household. | 2.39 | 1.208 | |
| Other household members make it impossible for me to reduce the amount of food wasted in my household. | 2.45 | 1.291 | |
| **Expiration dates knowledge** | | | **0.57** |
| There are differences between the "use-by" and "best-by" dates. | 3.78 | 1.296 | |
| The "use-by" date means that the food may present a health risk from that date and should not be consumed after that date. | 3.77 | 1.153 | |
| The "best-by" date indicates how long a product will retain its specific characteristics (e.g., yogurt should be creamy) if stored properly. Products may continue to be consumed after this date. | 3.44 | 1.164 | |

**Table A1.** *Cont.*

| | Mean | Std. Deviation | Cronbach's α |
|---|---|---|---|
| | Statistic | Statistic | Statistic |
| **Food storage knowledge** | | | **0.57** |
| Fruits excrete a gas during storage, which keeps vegetables fresh longer. Fruits and vegetables should therefore be stored together. | 2.64 | 1.123 | |
| Raw potatoes should not be stored in the refrigerator. | 3.81 | 1.206 | |
| Leftovers from warm meals should be cooled down before they are put in the refrigerator or freezer. | 4.49 | 0.966 | |
| **Identity of a good host and a good provider** | | | **0.64** |
| It would be embarrassing to me if my guests ate all the food, I had prepared for them. They would probably have liked to eat more. | 2.81 | 1.409 | |
| I regularly buy many fresh products although I know that not all of them will be eaten. | 2.65 | 1.179 | |
| I like to provide a large variety of foods at shared mealtimes so that everyone can have something he or she likes. | 3.11 | 1.205 | |
| I always have fresh products available to be prepared for unexpected guests or events (e.g., illness). | 2.52 | 1.149 | |
| When I am expecting guests, I like to buy more food than is necessary because I am a generous host. | 3.62 | 1.110 | |
| **Household planning habits** | | | **0.65** |
| When I have made a shopping list, I always keep strictly to it. | 2.85 | 1.242 | |
| I am a person who likes to plan things. | 3.77 | 1.112 | |
| Before I prepare food, I always consider precisely how much I need to prepare and what I will do with the leftovers. | 2.89 | 1.186 | |
| I always plan the meals in my household ahead and I keep to this plan. | 2.74 | 1.183 | |

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
