# Peer review of "Targeting Youths’ Intentions to Avoid Food Waste: Segmenting for Better Policymaking"

_agriculture, doi:10.3390/agriculture11040284_

Round 1

Reviewer 1 Report

General Comments:
This study considers interesting issue on food wastes which poses threat to economic and environmental sustainability. Overall the paper is well written and designed. However, I have few issues that need to be considered by the authors. 

Introduction
The introduction is well written and supported with recent literature.

Point 1: However, it is not clear how the study contributes to the existing literature. I encourage the authors to highlight the knowledge gap and contributions in narrowing this gap.

Section 2

Point 2: Section 2 appears to be more of conceptual framework than literature review. I would suggest that you change it to conceptual framework.

Point 3: The authors apply an appropriate theory to address the research questions. The TPB model has four main concepts, notably intention, attitude, perceived norms and behavioural control. However, the authors have included about 10 constructs. I am not convinced why the authors deviated from the traditional TPB model. Why did the authors included personal and financial attitude instead of general attitude towards wasting food? Why subjective norms and personal norms? This applies to all the variables included in the model. There should be strong justifications for including these variables in the proposed theory. Is the traditional TPB model is not adequate to predict intention not to waste food? If so it must be explained in detail and demonstrate the need to explain the theory.

Point 3: Another weakness of this paper is that did not explain how each of the construct was measured. Were they latent constructs or observed constructs? What were the reliability of the items used to measure those constructs?

Point 4: Data collection is part of the methodology. Also the data collection process need to be explained properly. Restructure the methodology as follows:

Methodology

Measurement of constructs

Data analysis

Results

The results should begin with descriptive results as presented in Table 1. In addition to that, the authors should present summary statistics of all the constructs concluded in the model. Afterwards, the authors should provide reliability score (Cronbach alpha) of the items used to use the constructs.

The next step then is the presentation of the model. Since, it is not indicated in the methodology how the constructs were measured. It is difficult for me to comprehend the results with its discussion.

Author Response

Dear Reviewer 1,

We would like to express our special thanks to you for taking your time to read our manuscript and for your valuable comments, which helped us improve the manuscript. Please find below responses to your comments.

Ad 1. There are 3 segments, we corrected this mistake throughout the study.

Ad 2. We corrected the order of variables and checked the consistency of variables.

Ad 3. We added an appendix with a list of variables and their reliability scale.

Ad 4. We corrected this table.

Ad 5. We demonstrated the differences and similarities of the models in the “Discussion” section.

Ad 6. We corrected the significance of the “Financial attitude” variable.

Ad 7. We included the missing references.

We hope that we addressed your comments accordingly.

Yours truly,

Ewelina M. Marek-Andrzejewska

Reviewer 2 Report

The study explores the factors influencing youth’s food waste intentions by applying the Theory of Planned Behavior (TPB) and divides the respondents into segments to analyze the different intentions in specific socioeconomic characteristics. Research findings indicate significant differences between different genders and the places of residence and conclude the usefulness of the TPB model as well as the segmenting approach. The background and motivation of the study were well-illustrated, and the paper has diversified policy recommendations to target each of the segments.

Suggestions for the study include:

  1. Are there 3 or 4 segments? Need to clarify and check again how many segments were divided in the study, and revise the research result.
  2. The order of variables for each hypothesis listed in Figure 1 seems randomly listed as H1, H2, H6, H7, H3, H10, H9, H5, H4, & H8. Therefore, it is confusing when trying to contra distinguish the hypothesis development part (i.e., section 2.1) to figure 1. Also, the consistency of variables should be checked, for example, H3 is “Health Attitudes” (line104, 114) or “Perceived health risk” (figure 1)? H8 is “Knowledge of food storage” (line 160) or “Storage” (figure 1)? H9 is “Household planning” or just “planning”?
  3. The paper has demonstrated the development of each hypothesis. However, the study does not disclose how these variables are measured. Questionnaire items are suggested to disclose so that readers can understand how these questions are designed for measurement.
  4. Data in Table 1 need to clarify the meanings. For example, the Median/Mean of the Gender is 1,00/1,28. What does that mean? Same as the Median/Mean of the educational level is 6,00/5,44. The Median/Mean of the Place of residence is 1,00/2.29.
  5. Although the research results have illustrated the findings in each proposed model respectively, comparisons of the differences between the general model and other segment models might bring about more comprehensive findings.
  6. Check lines 367-368 with line 294. Is or isn’t “Financial attitude” significant in the general model?
  7. Check the missing reference that has been marked in the paper (e.g., line 174, 246, 300, 353…..).

Author Response

Dear Reviewer 2,

We would like to express our special thanks to you for taking your time to read our manuscript and for your valuable comments, which helped us improve the manuscript. Please find below responses to your comments.

Ad 1. We clarified how the study contributes to the existing literature.

Ad 2. We changed the name of Section 2.

Ad 3. We provided an explanation on the need to include further variables in the model. We did an extensive literature review on the topic of food waste and noted that there were a number of variables, which were particularly important in this field of research.

Ad 3a. We added an appendix with a list of variables and their reliability scale.

Ad 4. We restructured the “Methodology” section, as suggested.

Ad 6. We added the “Descriptive results” section.

Ad 7. We included the missing references.

We hope that we addressed your comments accordingly.

Yours truly,

Ewelina M. Marek-Andrzejewska

Round 2

Reviewer 1 Report

All my concerns have been addressed sufficiently.

Best wishes